# Violence, self-harm and drug or alcohol misuse in adolescents admitted to hospitals in England for injury: a retrospective cohort study

Annie Herbert, Ruth Gilbert, Arturo González-Izquierdo, Leah Li

▶ Prepublication history and additional material is available. To view please visit the journal (http://dx.doi.org/10.1136/bmjopen-2014-006079).

Population, Policy & Practice Programme, University College London Institute of Child Health, London, UK

**Correspondence to**
Annie Herbert;
annie.herbert.12@ucl.ac.uk

## ABSTRACT

**Objectives:** Of adolescents in the general population in England, we aimed to determine (1) the proportion that has an emergency admission to hospital for injury related to adversity (violence, self-harm or drug or alcohol misuse) and (2) the risk of recurrent emergency admissions for injury in adolescents admitted with adversity-related injury compared with those admitted with accident-related injury only.

**Design:** We used longitudinally linked administrative hospital data (Hospital Episode Statistics) to identify participants aged 10–19 years with emergency admissions for injury (including day cases lasting more than 4 h) in England in 1998–2011. We used the Office for National Statistics mid-year estimates for population denominators.

**Results:** Approximately 4.3% (n=141 248) of adolescents in the general population (n=3 254 046) had one or more emergency admissions for adversity-related injury (girls 4.6%, boys 4.1%), accounting for 50% of all emergency admissions for injury in girls and 29.1% in boys. Admissions for self-harm or drug or alcohol misuse commonly occurred in the same girls and boys. Recurrent emergency admissions for injury were more common in adolescents with adversity-related injury (girls 17.3%, boys 16.5%) than in those with accident-related injury only (girls 4.7%, boys 7.4%), particularly for adolescents with adversity-related injury related to multiple types of adversity (girls 21.1%, boys 24.2%).

**Conclusions:** Hospital-based interventions should be developed to reduce the risk of future injury in adolescents admitted for adversity-related injury.

## INTRODUCTION

Many adolescents exposed to adversity such as violence, self-harm or drug or alcohol misuse use secondary health services,[1 2] often repetitively.[3 4] For example, in a self-report survey of participants aged 15–16 years in England, 12.6% of those who had self-harmed had presented to hospital.[2] It is also estimated that approximately one-third of patients attending

### Strengths and limitations of this study

- Hospital Episode Statistics (HES) captured data on all admissions to National Health Service hospitals in England at 10–19 years of age in this study's cohort.
- The longitudinal link between admissions for each individual in HES data allowed us to study the burden of multiple emergency admissions for injury over time.
- However, violence, self-harm and drug or alcohol misuse are not always recognised at an admission, or consistently recorded, and therefore this study's estimates of prevalence of adversity are likely to be underestimates.

a hospital in England for self-harm re-attend for self-harm in the following year.[4] Improved management of adolescents exposed to adversity could reduce risk of repetition as well as the burden on secondary care.[5–7]

An admission to hospital provides the 'teachable moment'.[8] That is, both adolescents and their families may be more likely to engage with an intervention than if they had received it elsewhere. Hospital-based interventions to reduce the risk of future harm could benefit these adolescents by reducing episodes of injury, and may reduce recurrent emergency (ie, acute or unplanned) admissions for injury.

To date, there is a lack of evidence on how different types of adversity-related injury occur in the same adolescents over time. In addition, policymakers and service providers need to know how many adolescents have an emergency admission to hospital for adversity-related injury, their characteristics and their specific rates of readmission if they are to be feasibly targeted for intervention.

In this study, we used administrative hospital data and the Office for National Statistics (ONS) mid-year population estimates to estimate the number of adolescents in the general

population who have ever had an emergency admission to hospital for injury. We then estimated the prevalence of emergency admissions for injury related to violence, self-harm and drug or alcohol misuse (alone and co-occurring) in the general population. Finally, we determined the risk of recurrent emergency admissions for injury in adolescents who had at least one admission between 10 and 19 years of age for adversity-related injury compared with adolescents only ever admitted for accidental injury during the same period.

## METHODS
### Study population
Using administrative data from all admissions to National Health Service hospitals in England (Hospital Episode Statistics (HES)) in 1998–2011,[9] we derived a retrospective cohort of adolescents who turned 10 years old in 1998–2002, who could be observed throughout adolescence until 19 years (see online supplementary table S1).[10] Each individual also had to have at least one emergency admission for injury between 10 and 19 years of age.

### Admission data
The Health and Social Care Information Centre provided pseudonymised data on hospital admissions, the use of which did not require Research Ethics Committee approval.[11] An admission is defined by the National Health Service as any hospital case lasting longer than 4 h, and so includes long day-cases as well as overnight stays. We analysed any hospital transfers or admissions within 1 day after discharge as the same admission, as previously described.[12] We used the variable for method of admission ('admimeth') to define an emergency admission. We used all International Classification of Diseases 10th Edition (ICD-10) diagnosis codes recorded during an admission to categorise admissions as being for injury related to adversity or an accident (see online supplementary table S2).[13]

### Types of injury and age at emergency admission
We defined an emergency admission for injury as being related to adversity, comprising violence (maltreatment/assault/undetermined causes of injury), self-harm, or drug or alcohol misuse, using mutually exclusive clusters of ICD-10 codes (see online supplementary table S2). Violence was defined by previously validated codes, which would trigger consideration of violence by carers, peers or strangers.[12 14 15] We defined self-harm using codes that mentioned either 'self-harm' or 'self-poisoning'. Drug or alcohol misuse was defined by codes that mentioned 'alcohol', 'drugs', 'noxious substance' or 'solvent'. We defined an injury as being related to accidents only if no adversity codes were present, but there were codes from the ICD-10 *Accidents* subchapter (V01-X59).[13]

We grouped age at each admission to reflect age of onset of puberty (10–14 years), age of sitting secondary school examinations (15–17 years) and the legal age for buying alcohol (18–19 years).[16–19]

### Classification of adolescents according to types of injury and age at emergency admissions
We classed adolescents into groups according to all of their emergency admissions for injury between 10 and 19 years of age. Adolescents were classed as belonging to the 'adversity' group (any adversity-related injury between 10 and 19 years of age), 'accidents-only' group (no adversity-related injury but one or more accident-related injuries) or 'other causes' group (no adversity-related or accident-related injuries) (see online supplementary figure S1). Among adolescents in the adversity group, we determined the proportion exposed to violence-related, self-harm-related and drug or alcohol misuse-related injury at age 10–19 years, respectively. We further classified the adversity group into seven mutually exclusive subgroups: violence only, self-harm only, drug or alcohol misuse only, violence and self-harm, violence and drug or alcohol misuse, self-harm and drug or alcohol misuse, and violence, self-harm and drug or alcohol misuse.

We also grouped adolescents as above, according to their emergency admissions for injury at 10–14 years only, 15–17 years only and 18–19 years only (see online supplementary figure S2).

### Population denominators
We used ONS mid-year population estimates to derive population denominators.[20 21] These data are freely available online, broken down by sex and year of age.

### Analyses
We estimated the proportion of adolescents in the general population who had an emergency admission for injury between 10 and 19 years of age. We used the number of adolescents in our derived retrospective cohort as the numerator and ONS mid-year estimates for participants aged 10 years in 1998–2002 as the population denominator. We then calculated these proportions by types of injury at 10–19 years of age (adversity-related (adversity group and seven mutually exclusive subgroups) and accidents only (accidents-only group)) and by age group, as described above.

We calculated the proportion of adolescents in the adversity group (and subgroups) and in the accidents-only group who had an emergency admission for injury twice and three or more times between 10 and 19 years of age. We also calculated the proportions of adolescents with two or three or more admissions of any type (including non-emergency and non-injury).

We reported all results separately for girls and boys since differences between girls and boys have been reported for prevalence of adversity in the general population.[1 22–24] We calculated 95% CIs for all proportions but did not present them here as they were all too narrow to convey any useful information (within one

unit of the sample estimate). Analyses were carried out in StataSE V.12.

## RESULTS
There were 1 033 702 adolescents in HES admissions data in 1998–2011, of which 402 916 formed the study cohort (462 476 emergency admission for injury when considering multiple presentations from the same adolescent, 802 682 admissions of any type (including non-emergency and non-injury, 662 727 of which were overnight stays)) (table 1), representing 12.4% (402 916/3 254 046) of the adolescent population. Twice as many boys as girls had an emergency admission for injury during adolescence (144 158/1 588 942 girls in the population (8.7%); 258 503/1 665 104 boys (16.3%)).

### Types of injury and age at emergency admission
One-third of the cohort (141 248, 4.3% of the population) had a record of an emergency admission for adversity-related injury between 10 and 19 years of age (the adversity group; 157 004 emergency admissions for adversity-related injury in total) (table 1), with similar rates between sexes (72 805, 4.6% girls in the population; 68 403, 4.1% boys).

The remaining two-thirds of the cohort (261 668, 8.1% of the adolescent population) had emergency admissions for injury which were never related to adversity (table 1). Among these adolescents, 233 907 (89.4%) had an accident-related injury (the accidents-only group, 7.2% of the population) and 27 761 (10.6%) had no accident-related injury (other causes group, 0.9% of the population). A high proportion of the other causes group were affected by a chronic condition[i] between 10 and 19 years of age (11 221/27 761, 40.4%), compared with the adversity group (45 321/141 248, 32.1%) or accidents-only group (49 434/233 907, 21.1%).

Proportions of adolescents in the general population and within individual age groups by adversity (and subgroups), accidents-only, and other causes groups are provided in online supplementary table S3.

### Types of adversity-related injury
Among adolescents in the adversity group (girls 72 805, boys 68 403) (figure 1), the most common type of adversity was drug or alcohol misuse (girls 91.5%, boys 60%). A higher proportion of boys than girls were exposed to violence (girls 8.5%, boys 47.6%), but a higher proportion of girls than boys were exposed to self-harm (girls 74.6%; boys 32.9%).

Girls in the adversity group were most likely to be exposed to multiple types of adversity between 10 and 19 years of age (69.2%+2.0%+1.2%+0.2%=72.6%; figure 1), especially self-harm and drug or alcohol misuse (69.2% of the entire adversity group, ie, most of

---

[i]Defined by ICD-10 codes (see online supplementary table S2).

the 72.6%). Fewer boys in the adversity group were exposed to multiple types of adversity (38.4%), the most common combination also being self-harm and drug or alcohol misuse (24.8%).

For most of the adolescents who were exposed to multiple types of adversity, the combination of types was recorded at the same admission. For example, among the 130 adolescent girls who were exposed to violence and self-harm between 10 and 19 years of age (table 2), 64.6% had both violence and self-harm codes present simultaneously in at least one emergency admission for injury (violence and drug or alcohol misuse 78.8%, self-harm and drug or alcohol misuse 99.7%, violence, self-harm and drug or alcohol misuse 33.9%; boys: violence and self-harm 40.1%, violence and drug or alcohol misuse 84.1%, self-harm and drug or alcohol misuse 99.1%, violence, self-harm and drug or alcohol misuse 20.0%) (data not shown).

### Emergency readmissions for injury
Adolescent girls in the adversity group (50.5% of all girls in the cohort) accounted for 50% of the total number of emergency admissions for injury coming from girls (data not shown), compared with girls in the accidents-only group (41.3% of all girls) who accounted for 36.6%. Boys in the adversity group (26.2% of all boys in the cohort) accounted for 29.1% compared with 65% contributed by boys in the accidents-only group (67.7% of all boys).

More adolescents in the adversity group were readmitted for injury (ie, had two or more emergency admissions for injury) between 10 and 19 years of age (girls 17.3%, boys 16.5%; figure 2) than in the accidents-only group (girls 4.7%; boys 7.4%). Among adolescents admitted for injuries related to multiple types of adversity (table 2), the proportion readmitted was even higher (multiple types: girls 21.1%, boys 24.2%; single type: girls 7.2%, boys 10.1%).

Similarly, a higher proportion of adolescents in the adversity group had two more admissions of any type (including non-emergency and non-injury) between 10 and 19 years of age (girls 46.2%, boys 35.2%; table 2) compared with adolescents in the accidents-only group (girls 33.4%, boys 28.5%). This proportion was even higher for adolescents in the adversity group who were admitted with multiple types of adversity (multiple types: girls 49.0%, boys 42.5%; single type: girls 38.8%, boys 25.5%).

## DISCUSSION
More than 1 in 20 adolescents in England had at least one emergency admission for adversity-related injury between 10 and 19 years of age. These adolescents accounted for a third of all adolescents with emergency admissions for injury and for a disproportionate number of readmissions for injury, particularly adolescents admitted with multiple types of adversity-related injury. Targeting adolescents admitted with adversity-related

**Table 1** Characteristics of adolescents whose entire 10 years of adolescence (ages 10–19) occurred in 1998–2011

| Characteristics | Adolescent population* Total | Adolescents with emergency admission(s) for injury between 10 and 19 years of age, n (row %) | | | |
| --- | --- | --- | --- | --- | --- |
| | | Total | Adversity | Accidents only | Other causes |
| All | 3 254 046 | 402 916 (100.0) | 141 248 (35.1) | 233 907 (58.1) | 27 761 (6.9) |
| Age† | | | | | |
| Girls (years) | 1 588 942 | 144 158 (100.0) | 72 805 (50.5) | 59 528 (41.3) | 11 888 (8.2) |
| 10–14 | | 65 208 (100.0) | 23 178 (35.5) | 37 388 (57.3) | 4642 (7.1) |
| 15–17 | | 48 286 (100.0) | 31 573 (65.4) | 12 922 (26.8) | 3791 (7.9) |
| 18–19 | | 30 664 (100.0) | 18 054 (58.9) | 9155 (29.9) | 3455 (11.3) |
| Boys (years) | 1 665 104 | 258 503 (100.0) | 68 403 (26.5) | 174 267 (67.4) | 15 833 (6.1) |
| 10–14 | | 121 821 (100.0) | 17 667 (14.5) | 97 478 (80.0) | 6676 (5.5) |
| 15–17 | | 79 223 (100.0) | 25 014 (31.6) | 49 345 (62.3) | 4864 (6.1) |
| 18–19 | | 57 459 (100.0) | 25 722 (44.8) | 27 444 (47.8) | 4293 (7.5) |
| Missing (sex) | | 255 (100.0) | 40 (15.7) | 175 (68.6) | 40 (15.7) |

*Based on the Office for National Statistics (ONS) mid-year England statistics for participants aged 10 years in 1998–2002.[20]
†At first emergency admission for injury.

injury could reduce their risk of future harm, the rate of readmissions to hospital and healthcare costs.[25]

Longitudinally linked admissions allowed us to study the entire 10 years of adolescence in 402 916 individuals. We were able to distinguish between types of adversity that co-occurred during adolescence or at the same admission, and to study readmissions. One weakness of this study was our reliance on diagnostic codes recorded in administrative data. Violence by carers, which could be coded under maltreatment, and drug or alcohol misuse have been shown to be under-recorded using ICD-10,[26 27] but false positives are rare.[15] To address under-recording, we used what we considered to be sensitive clusters of codes for adversity. Other factors related to recording or coding practices,[12 14 15 27 28] for example, new guidelines for defining maltreatment,[14] can also affect ascertainment. Owing to the relative insensitivity but good

specificity of the coding clusters, some adolescents who were classified in the accidents-only group may in fact belong to the adversity group, but did not have their adversity recognised or recorded. Consequently, our prevalence estimates of admission for different types of adversity-related injury are likely to provide a lower bound for the true prevalence. Further, as adolescents exposed to adversity who attended the accident and emergency (A&E) department were not necessarily admitted, our prevalence estimates represent adolescents at the severe end of the adversity spectrum. Such analyses of A&E data are limited by the quality of these data in England (available since 2007) and the resulting problems with identifying reasons for presentation and accurately linking individuals to long-term outcomes.[29]

Our prevalence estimates of admission for injury related to individual types of adversity from the general

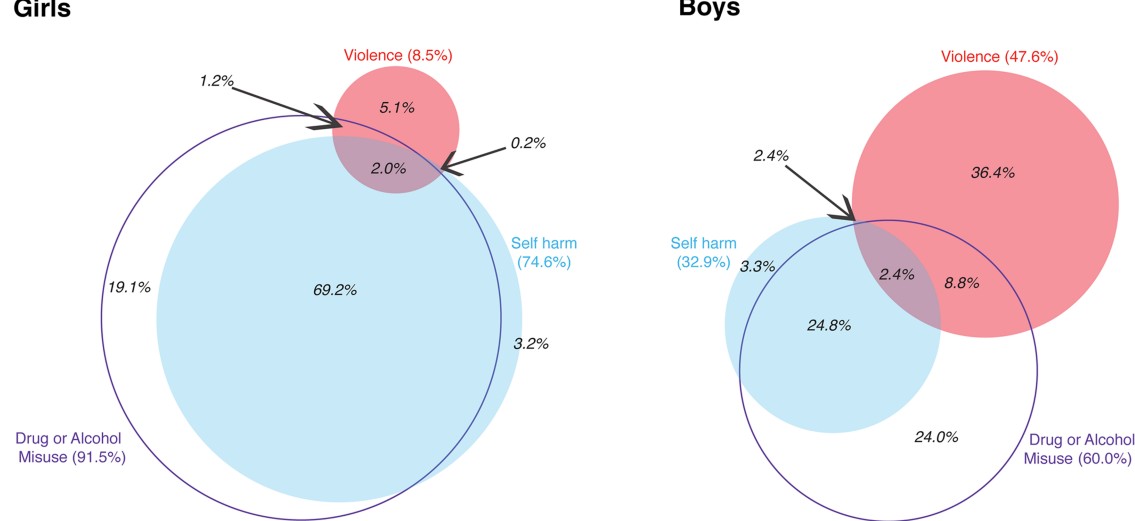

**Figure 1** Number (%) of adolescents with adversity-related injury, by types of adversity between 10 and 19 years of age and sex. Each adolescent classified by all adversity recorded at any emergency admission(s) for injury between 10 and 19 years of age.

**Table 2** Proportion of adolescents with 1, 2 or 3+ emergency admission(s) for injury or 1, 2 or 3+ admission(s) of any type, by types of adversity between 10 and 19 years of age*

| Adolescent group* | Girls (%) | | | | | | | Boys (%) | | | | | | |
| | Number of girls | Emergency admission(s) for injury | | | Admission(s) of any type | | | Number of boys | Emergency admission(s) for injury | | | Admission(s) of any type | | |
| | | 1 | 2 | 3+ | 1 | 2 | 3+ | | 1 | 2 | 3+ | 1 | 2 | 3+ |
|---|---|---|---|---|---|---|---|---|---|---|---|---|---|---|
| All | 144 158 | 88.6 | 8.3 | 3.1 | 57.6 | 20.5 | 22.0 | 258 503 | 90.3 | 8.0 | 1.8 | 68.5 | 18.7 | 12.9 |
| Adversity | 72 805 | 82.7 | 12.0 | 5.3 | 53.8 | 21.1 | 25.1 | 68 403 | 83.5 | 12.4 | 4.1 | 64.8 | 19.4 | 15.8 |
| Any violence | 6211 | 77.2 | 13.9 | 8.9 | 49.1 | 20.7 | 30.1 | 32 799 | 83.2 | 12.8 | 4.0 | 65.6 | 19.6 | 14.8 |
| Any self-harm | 54 315 | 79.3 | 13.9 | 6.8 | 51.2 | 21.5 | 27.3 | 21 087 | 76.7 | 15.8 | 7.5 | 57.0 | 20.7 | 22.3 |
| Any drug or alcohol misuse | 66 645 | 81.9 | 12.5 | 5.6 | 53.6 | 21.1 | 25.3 | 41 014 | 81.1 | 13.6 | 5.3 | 62.9 | 19.5 | 17.6 |
| Single adversity | 19 924 | 92.8 | 6.2 | 1.0 | 61.2 | 19.8 | 19.0 | 43 563 | 71.3 | 8.3 | 1.8 | 55.8 | 15.2 | 10.3 |
| Violence only | 3734 | 92.4 | 6.3 | 1.3 | 58.1 | 20.4 | 21.6 | 24 912 | 87.1 | 10.6 | 2.2 | 68.3 | 19.2 | 12.6 |
| Self-harm only | 2296 | 90.3 | 8.1 | 1.6 | 54.4 | 20.7 | 24.9 | 2260 | 87.3 | 10.4 | 2.3 | 62.3 | 19.8 | 17.9 |
| Drug or alcohol misuse only | 13 894 | 93.3 | 5.9 | 0.8 | 63.2 | 19.5 | 17.4 | 16 391 | 88.5 | 9.5 | 2.1 | 69.9 | 17.9 | 12.2 |
| Multiple adversity | 52 881 | 78.9 | 14.2 | 6.9 | 51.0 | 21.6 | 27.4 | 24 840 | 77.3 | 16.7 | 7.5 | 59.0 | 21.0 | 21.5 |
| V+SH | 130 | 70.0 | 20.8 | 9.2 | 42.3 | 26.2 | 31.5 | 217 | 41.6 | 15.4 | 6.1 | 32.8 | 16.0 | 14.2 |
| V+DA | 862 | 81.9 | 15.1 | 3.0 | 52.2 | 22.9 | 24.9 | 6013 | 84.6 | 17.5 | 5.4 | 68.4 | 21.8 | 17.3 |
| SH+DA | 50 404 | 80.1 | 13.7 | 6.3 | 51.8 | 21.6 | 26.6 | 16 953 | 86.7 | 16.4 | 6.9 | 64.6 | 22.7 | 22.9 |
| V+SH+DA | 1485 | 36.8 | 31.6 | 31.5 | 25.5 | 19.9 | 54.5 | 1657 | 22.0 | 16.4 | 14.1 | 17.9 | 11.7 | 22.8 |
| No adversity | 71 353 | 94.7 | 4.5 | 0.8 | 61.4 | 19.9 | 18.7 | 190 100 | 92.4 | 6.3 | 0.9 | 69.5 | 18.3 | 11.8 |
| Accidents only | 59 465 | 95.3 | 4.1 | 0.6 | 66.5 | 18.7 | 14.7 | 174 267 | 92.3 | 6.5 | 0.9 | 71.1 | 18.0 | 10.5 |
| Other causes | 11 888 | 91.5 | 6.3 | 2.1 | 35.5 | 25.8 | 38.8 | 15 833 | 93.4 | 5.0 | 1.6 | 51.5 | 22.1 | 26.3 |

*Each adolescent classified by all adversity/accidents seen at any emergency admission(s) for injury between 10 and 19 years of age.
DA, drug or alcohol misuse; SH, self-harm; V, violence.

adolescent population are consistent with previous reports for emergency admissions for assault-related injury in 2004–2009 and for all admissions (emergency and non-emergency) for self-harm and drug or alcohol misuse.[1][2][24][30] Previous studies have reported higher rates of drug or alcohol misuse in boys than in girls in the general adolescent population.[24] We found higher rates in girls. This difference could indicate that girls exposed to drug or alcohol misuse are more likely to be injured, to present to hospital, or to be admitted after a hospital presentation, than boys.

Our estimated rates of readmission of any type (including non-emergency and non-injury) for violence (girls 50.8%, boys 34.4%) (table 2) and self-harm (girls 48.8%, boys 43.0%) were higher than previously reported (11% for violence, 33% for self-harm).[3][4]

**Figure 2** Number of adolescents with 1, 2 and 3 or more emergency admission(s) for injury between 10 and 19 years of age, by types of injury between 10 and 19 years of age and sex. Percentages are of adolescents who have two or more emergency admissions for injury between 10 and 19 years of age. Each adolescent was classified by all adversity/accidents recorded at any emergency admission(s) for injury.

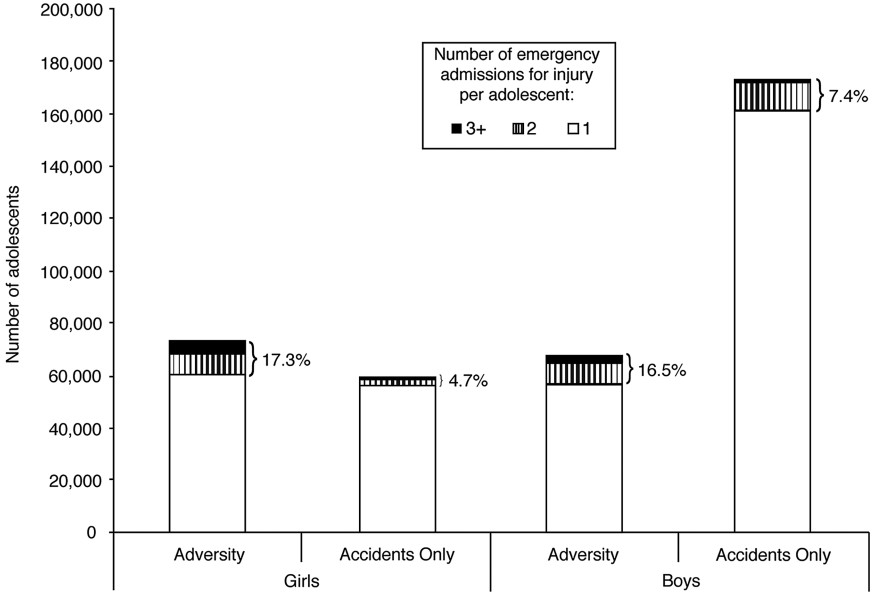

These discrepancies are likely because we considered the whole 10 years of adolescence and readmission of any type, whereas previous studies looked at re-attendance up to the following year and for the same type of adversity-related injury.[3 4]

The results of this study should inform policy initiatives and national guidelines. First, a substantial proportion of adolescents are affected by adversity and they account for a large proportion (29.1–50.0%) of all emergency admissions to hospital for injury in this age group. Second, we show the large burden of injury admission for all three types of adversity, yet there are currently no national clinical guidelines for managing cases of violence, other than responding to violence by caregivers.[31] Finally, these results show that adolescents often present with multiple types of adversity (especially in girls), even though guidelines exist only for managing individual problems.[32–34]

In addition, policymakers need to be aware of the widely varying aetiological pathways to admission with adversity-related injury. Our approach to defining this group of adolescents is not designed to reflect the complexity or severity of these cases. For example, admission for multiple types of adversity-related injury is a poor proxy indicator of severity. Effective interventions will need to be tailored to the individual based on specialist clinical assessment. However, all three types of adversity are likely to reflect a combination of underlying psychosocial need and environmental and social stressors.[35]

Hospital interventions may reduce the risk of future harm, including the incidence of other types of harm not seen in hospital, for example, further adversity-related injury not leading to admission. Further research using linked data from healthcare sectors such as A&E could shed light on the overall burden of adversity-related injury on hospitals. Although these data have limited quality in England, longitudinally linked data sets in other countries could provide insights into long-term outcomes for this vulnerable group of adolescents.

**Contributors** AH conceived and designed the study, analysed and interpreted the data, drafted the article, revised it critically for important intellectual content and approved the final version to be published. LL conceived and designed the study, interpreted the data, revised the article critically for important intellectual content and approved the final version to be published. AG-I and RG conceived and designed the study, acquired and interpreted the data, revised the article critically for important intellectual content and approved the final version to be published.

**Funding** AH was supported by the Policy Research Unit in the Health of Children, Young People and Families, which is funded by the Department of Health Policy Research Programme (grant reference number: 109/0001). This report is an independent research commissioned by the Department of Health. She is also supported by the University College London IMPACT studentship. RG is supported by awards establishing the Farr Institute of Health Informatics Research at University College London Partners from the Medical Research Council and a consortium of funders (MR/K006584/1).

**Competing interests** None.

**Ethics approval** Both Hospital Episode Statistics (HES) admissions data and Office for National Statistics (ONS) mid-year population estimates are derived from routinely collected administrative data. HES data were pseudonymised before we received them, and therefore we did not require Research Ethics Committee approval.

**Provenance and peer review** Not commissioned; externally peer reviewed.

**Data sharing statement** No additional data on HES are available. ONS mid-year population estimates may be accessed freely online: http://www.ons.gov.uk/ons/publications/all-releases.html?definition=tcm%3A77-22371

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
