## [Reviewer comments · BMJ Open]

Some articles will have been accepted based in part or entirely on reviews undertaken for other BMJ Group journals. These will be reproduced where possible.

ARTICLE DETAILS

TITLE (PROVISIONAL)	Violence, self-harm and drug or alcohol misuse in adolescents admitted to hospitals in England for injury: a retrospective cohort study
AUTHORS	Herbert, Anne; Gilbert, Ruth; Gonzalez-Izquierdo, Arturo; Li, Leah

VERSION 1 - REVIEW

REVIEWER	David Cottrell University of Leeds, London Ruth Gilbert and I are currently collaborating on a grant application in a related field to the content of this paper.
REVIEW RETURNED	21-Jul-2014

GENERAL COMMENTS	I found this an interesting submission that has some important things to say. The authors use hospital administrative data and national population statistics to explore factors associated with emergency admission in teenagers. The descriptions of the scale of adversity in this population and the prevalence of emergency admission are of relevance in themselves but the evidence presented about the links between the two will be of interest to all those concerned with child health and with emergency departments. The authors have done a good job of presenting what they did clearly and the supplementary tables and figures make it possible to follow the methods used in detail. However, it is still a difficult read with a plethora of acronyms, abbreviations and strings of numbers in brackets breaking up the text. Personally I would prefer to see a number of the acronyms such as AdvRI and AccRI written out in full as 'adversity related' and 'accident only' each time. My main concern with the paper is its use of language and the definition of terms used. It would be helpful to confirm that 'admission' refers to transfer from an ED to a ward and that the paper is not referring to those teenagers who present to A&E in the main text rather than in a supplementary table. It would be interesting to know, in this context, how long these admissions were. I do not like the term 'victimisation'. Supplementary table 2 provides a helpful list of how this was defined. I can see that evidence of actual abuse or of being assaulted might be styled 'victimisation' but this category includes (leaving out rather bizarre entries like 'neonatal withdrawal symptoms from maternal use of drugs' – p33,
---

	line 28) thing like 'inadequate housing' or 'family history of mental and behavioural disorder'. I can see that these may represent adversity for the young person but not that they represent victimisation! The term is therefore potentially misleading and in my opinion needs changing. The authors seem also to have assumed that any evidence of alcohol in an under 18 yr old is an indication of adversity. I appreciate the difficulty in knowing what relevance alcohol might play in the admission when using existing data but this does make interpretation of finding difficult. What did the authors do about evidence of alcohol use in the small numbers who were over 18yrs. Was this ignored or was some analysis conducted that used the data to decide if the use was misuse, and if so what was this and could it be applied to older teenagers under 18yrs? Notwithstanding these comments, which I think could be addressed relatively easily, I found this an interesting paper with valuable conclusions with the potential to influence practice.
--	--

REVIEWER	Simon Moore Cardiff University, UK
REVIEW RETURNED	29-Jul-2014

GENERAL COMMENTS	A reasonably useful paper - I particularly found the recurrent admission data of interest, it really brings home the need for interventions in this group. But ultimately all that is done is count the number of adolescents admitted to hospital and divide by population estimates, with a few descriptive stats thrown in for good measure. A lot more could be done with these data. For example modelling and simulations might give interesting insights in the effectiveness of an intervention that reduces readmission by a certain percentage. Could usefully justify why the 10-19 year group was chosen and why ages were grouped in 10-14, 15-17 and 18-19 for analysis - seems like this grouping reduces the information available. Table 1 - data is presented for "Adolescents whose entire ten years of adolescence (ages 10-19) occurred in 1998-2011" which makes sense. But what about the denominator? Is the cited 3,254,046 also for all adolescents 1998-2011? Also, surely you would need births and deaths data to approach an accurate figure? It would be nice to know the distribution of admissions, that is how many had one AdvRI, two, three... and so on. At the moment Table 2 stops at "3+". Might be nice to have cumulative admissions in a graph for each group. This paper focuses only on admissions the very severe end of the spectrum. A great number of youngsters will attend, be treated but not admitted. It is important that this paper acknowledges this. While not as costly there are still impacts on services. Particularly if making the case that interventions are required in this group. Minor comments: Page 2 - line 40. Numbers are hard to read
--

	Supplementary Table 1 does not seem to add very much
--	--

REVIEWER	Peter Sidebotham Warwick Medical School, UK
REVIEW RETURNED	26-Nov-2014

GENERAL COMMENTS	This is a tidy study using secondary national data to examine adversity-related injury admissions among adolescents in England. The messages are straightforward and important:  - adversity-related injury is prevalent in the English adolescent population, accounting for 60% of all emergency admissions for injury in girls and 30% in boys; - co-morbidity is common, with many of this group suffering multiple adversities; - recurrence is more common in this group than among those with accident-related injury. These are extremely important messages with implications for practitioners and policy makers/public health. The study has been well designed, and the authors have been thorough in describing their approach to coding adversity-related injury. The methods are clearly described and appropriate. I was uncertain that the inclusion of coding for and analysing the data in relation to chronic conditions really added anything to the paper. Unless you are making any hypotheses in relation to chronic conditions, or intend to explore this in depth, I suspect it adds an unnecessary layer of complexity. I would exclude this component of the analysis in the interests of keeping the key messages clear and simple. The results are clearly presented and easy to follow. The tables are clear (as above, the row on chronic conditions could be omitted). The discussion is well presented, thorough and appropriate in the light of your results and the wider literature. You discuss the limitations of this dataset. You draw out appropriate policy implications. I felt figure 1 was unnecessarily complex. Why not just present as 2 straightforward Venn Diagrams? The data in the boxes on the right could simply be incorporated in the relevant fields of the Venn diagram. Minor comments:  1. p6, line 93 - you need to be consistent in hyphenating adversity-related 2. p6, line 95 - need to know 3. p8, line 128 - as being related to 4. p9, line 134 - I presume the comment 'and excluded 'undetermined intent'.' means this was excluded from the self-harm category, but included under assault. This could be clarified, as my first thought on reading it was that you were excluding it entirely, which would not be appropriate as it implies some adversity. 5. p16, line 233 - adolescents who were exposed to
--

	6. In the supplementary table (p33, line 28), surely neonatal withdrawal symptoms could be excluded as have other neonatal conditions. 7. p36, line 34 - should X65 be included in both SH and DA? Similarly X66 (p38 line 5) 8. p38, line 29 - I am unconvinced by the exclusion of Z86.4 because it includes tobacco. Surely the inclusion of alcohol and drugs here makes its inclusion important? This may need further justification. 9. p40, line 6. In the title for this table, you need to specify that the population prevalence is per 100,000 (I presume)
--	---

REVIEWER	Kirsten Vallmuur Queensland University of Technology Centre for Accident Research and Road Safety Queensland
REVIEW RETURNED	04-Dec-2014

GENERAL COMMENTS	This paper describes admissions for adversity-related problems and injury in adolescents (10-19 years) in England using longitudinally linked retrospective hospital admission data in England. It is a very well written, innovative and interesting paper using a well constructed methodology, which addresses a significant problem of concern to the health sector and larger community. The comments I have are all focused on providing greater clarity in the description to assist the readers understanding, as well as a few minor editorial suggestions. Firstly, my most substantial recommendation which affects the wording used throughout the paper is around the use of the term 'injury'. While I note that the Supplementary Table 2 states that for a case to be considered an injury it would have an S or T code from the ICD in the first episode of admission, this is the only place where this restriction is indicated; the paper itself refers the reader to the adversity codes and accident codes on page 8 line 128-129 "we defined an emergency admission for injury as being for related to adversity....." (note the grammar needs correcting in this sentence on review), and hence it is unclear to me whether the case cohort is required to have S or T codes to be included OR whether it can have S or T codes or any of the adversity codes. This needs to be made clearer and if it is the latter (ie S OR T OR Adversity codes) I have some thoughts on the terminology used throughout the paper. If using the operational definition of 'injury' which is largely employed in the injury field, technically injuries are only those coded to Chapter 19 and/or Chapter 20 of the ICD-10. Even the more theoretical definitions of injury which have been put forward over the years (such as the World Health Organisation definition in Holder Y, Peden M, Krug E et al (Eds). Injury surveillance guidelines. Geneva, World Health Organization, 2001) would not cover the entirety of the concepts which are included within the 'adversity' category. I agree with the breadth of the adversity concept, which I think is an inclusive and comprehensive list of relevant codes, however I think that there could be a better use of terminology than 'injury' in relation to the proposed 'adversity' category. One approach could be to use the phrases:  • 'adversity-related injuries and social problems' (or simply 'adversity-related problems') instead of 'adversity-related injuries';
--

- 'adversity and/or injury emergency admissions' instead of 'emergency admissions for injury'; and
- 'problems related to adversity' instead of 'injury related to adversity'.

I agree with the use of the term 'accident-related injury' for the categories referring to the cases with a code in the unintentional injury code range without the adversity codes, as these are within the scope of what is an 'injury'.

The other suggestions I have for improving clarity are as follows:

- TITLE: The title should reflect the change suggested in previous paragraph – suggested wording '....in England for adversity-related problems and injury.....';
- ABSTRACT, DESIGN: Include the fact that data longitudinal and linked as this makes it unique – i.e. 'we used longitudinally linked hospital administrative data'
- METHODS, ADMISSION DATA: Last sentence indicates the codes for 'other causes' is in the table but they aren't currently and there needs to be either a broad range provided or a statement to say 'all other codes in ICD other than those listed above'.
- METHODS, TYPES OF INJURY: Suggest adding to '(maltreatment/assault)' line 129 '(maltreatment/assault/social problems)' to give the reader an early indication that the adversity category is broad.
- METHODS, TYPES OF INJURY: Age ranges in line 142 are overlapping and should be (15-17 years) and (18+ years).
- RESULTS, FIRST PARAGRAPH: there were 402916 cases but 462476 emergency admissions? Does this include the multiple presentations for some of the cases? If so this needs to be made more explicit.
- RESULTS, TABLE 1: It wasn't clear till I found the footnote that the table referred to the first emergency admission and I would suggest including this in the column heading and maybe even the table title to make it more transparent. Also, I don't see the need to repeat the column headings and 'All' row again in the second row from the bottom and I would suggest just including the chronic conditions in a consecutive row after the Missing (sex) row.
- RESULTS, TYPES OF ADVERSITY-RELATED INJURY: Line 230, add 61.7% 'of all of the AdvRI group' to make it clear what the denominator is, and similarly for line 232 after 22.7%.
- RESULTS, TYPES OF ADVERSITY-RELATED INJURY: Line 233, should read 'for most of the adolescents who were exposed'.
- RESULTS, TYPES OF ADVERSITY-RELATED INJURY: Line 237, should read 'simultaneously *in* at least'.
- RESULTS, TABLE 2: Is it possible to make the columns AdvRI, AccRI, Other instead of Emergency admissions and Admission of any type? These latter categories don't mean much in the context of the article whereas having some indication of how many cases had second and third representations for adversity related problems or Injury related problems for each group using all of the same row headings would be of interest to get a more complete understanding of the trajectory of these patients.
- DISCUSSION: In regards to Line 300 where the authors

	indicate a higher rate of drug/alcohol misuse in girls in their study which they state is different to other studies, the study recently published online in DAR on ED injury presentations in youth and the level of drug/alcohol involvement found the same finding (See http://www.ncbi.nlm.nih.gov/pubmed/25303680).  • DISCUSSION: Line 304 – are those with adversity presentations more likely to be readmitted if they reappear even if the injury is minor so as to remove them from dangerous or suboptimal home environments (homeless, domestic violence etc) for a short stint, and does this impact on the findings in relation to readmissions? • DISCUSSION: Line 315 – the authors mention there are no clinical guidelines for assault, so does this mean there are clinical guidelines for self harm and alcohol/drug admissions? If not then these need to be added to the previous sentence in the manuscript. • DISCUSSION: Needs a concluding paragraph to tie up the paper, and this could briefly mention what the international significance of this project is (i.e. ability to replicate methods in other countries that have linked data etc).
--	---

VERSION 1 – AUTHOR RESPONSE

Reviewer: 1

Reviewer Name David Cottrell

Institution and Country University of Leeds, London

I found this an interesting submission that has some important things to say. The authors use hospital administrative data and national population statistics to explore factors associated with emergency admission in teenagers.

The descriptions of the scale of adversity in this population and the prevalence of emergency admission are of relevance in themselves but the evidence presented about the links between the two will be of interest to all those concerned with child health and with emergency departments.

1. The authors have done a good job of presenting what they did clearly and the supplementary tables and figures make it possible to follow the methods used in detail.

However, it is still a difficult read with a plethora of acronyms, abbreviations and strings of numbers in brackets breaking up the text. Personally I would prefer to see a number of the acronyms such as AdvRI and AccRI written out in full as 'adversity related' and 'accident only' each time.

Response:

We have removed 'AdvRI' and 'AccRI' from the manuscript revision throughout. We left in the acronyms for ONS and HES as these are frequently used in the literature. We have kept the acronyms for combinations of types of adversity, e.g., V+SH, in the tables and figures, as we feel that this is still simpler to follow than writing out the full combinations for all.

2. My main concern with the paper is its use of language and the definition of terms used. It would be helpful to confirm that 'admission' refers to transfer from an ED to a ward and that the paper is not referring to those teenagers who present to A&E in the main text rather than in a supplementary table. It would be interesting to know, in this context, how long these admissions were.

Response:

Both in the main text and Supplementary Table 2, admission refers to a stay lasting longer than four hours. Thus admission could include a long day case, or an overnight stay. We have now made this clearer in the Abstract (line 36), Methods (lines 123-125) and Results (lines 191-192) which should now marry up with the definition given in Supplementary Table 2.

For the cohort 29,223/462,476 (6.3%) of all emergency admissions for injury and 139,955/802,682 (17.4%) of all admissions (of any type) at 10-19 years old were long day cases. Therefore, including long day cases in the definition of admission should not have a substantial effect on whether individuals are classed as Adversity or Accidents Only, or on the associated burden of admissions between groups.

3. I do not like the term 'victimisation'. Supplementary table 2 provides a helpful list of how this was defined. I can see that evidence of actual abuse or of being assaulted might be styled 'victimisation' but this category includes (leaving out rather bizarre entries like 'neonatal withdrawal symptoms from maternal use of drugs' – p33, line 28) things like 'inadequate housing' or 'family history of mental and behavioural disorder'. I can see that these may represent adversity for the young person but not that they represent victimisation! The term is therefore potentially misleading and in my opinion needs changing.

Response:

We used 'neonatal withdrawal symptoms from maternal use of drugs' in a previous paper on victimisation in a much younger age group and think it has snuck its way in. This has been removed from our analysis as the other neonatal conditions have been.

We have now removed all codes in the section 'Adverse Social Circumstances' from the victimisation coding cluster, and refer to 'victimisation' as 'violence' so there is less ambiguity about what this cluster represents. This means that 1,677/47,713 (3.5%) adolescents who were originally defined as having a record of victimisation at 10-19 years old and 1,017/140,512 (0.7%) who were originally in the Adversity group are no longer defined as such. All tables, figures and results in the main text have been altered according to the new definition of violence. Proportions and associations in the revised tables and figures do not alter substantially because of the new violence coding cluster.

4. The authors seem also to have assumed that any evidence of alcohol in an under 18 yr old is an indication of adversity. I appreciate the difficulty in knowing what relevance alcohol might play in the admission when using existing data but this does make interpretation of findings difficult. What did the authors do about evidence of alcohol use in the small numbers who were over 18yrs. Was this ignored or was some analysis conducted that used the data to decide if the use was misuse, and if so what was this and could it be applied to older teenagers under 18yrs?

Response:

18-19 year olds who showed evidence of alcohol use (represented by code Z72.1), but not evidence of harmful or dependent drinking were not categorised as having been exposed to drug or alcohol misuse.

We have decided that any evidence of alcohol use (Z72.1) and 'evidence of alcohol involvement determined by level of intoxication' (Y91) should be extended to 18-19 year olds for defining misuse. Though it is difficult to know the level of alcohol that this represents, even if the adolescent is at the legal age of drinking, the fact that this alcohol use is coupled with an emergency admission for injury and that the clinician has felt the need to record this alcohol involvement, means that these cases are likely to indicate alcohol misuse. This means that the number of adolescents in the drug or alcohol misuse sub-group has increased by 2%, from 105,519 to 107,686. All tables, figures and results in the main text have been altered according to the new definition of drug or alcohol misuse, and do not

substantially differ to the original results.

Notwithstanding these comments, which I think could be addressed relatively easily, I found this an interesting paper with valuable conclusions with the potential to influence practice.

Reviewer: 2

Reviewer Name Simon Moore

Institution and Country Cardiff University, UK

1. A reasonably useful paper - I particularly found the recurrent admission data of interest, it really brings home the need for interventions in this group. But ultimately all that is done is count the number of adolescents admitted to hospital and divide by population estimates, with a few descriptive stats thrown in for good measure. A lot more could be done with these data. For example modelling and simulations might give interesting insights in the effectiveness of an intervention that reduces readmission by a certain percentage.

Response:

Thank you for your comments. We agree that these data have a lot of a potential. This is the first in a series of papers which will shed light on how much difference targeted interventions could make.

2. Could usefully justify why the 10-19 year group was chosen and why ages were grouped in 10-14, 15-17 and 18-19 for analysis - seems like this grouping reduces the information available.

Response:

We chose 10-19 years old as this is widely adopted as the definition of an adolescent in the literature (Lancet series on Adolescents and World Health Organisation), and reflects a period when young people experiment with risk-taking behaviours, and so provide opportunity for intervention which could affect the rest of the adolescent's life course.(1) We presented the prevalence of combinations of adversity types by age since we know that the prevalence of hospital admissions for violence changes with age, and that the prevalence of self-harm and drug or alcohol misuse in the general adolescent population changes with age.(2-4) Combinations of adversity types are difficult to present by individual years of age because the data would become sparse and difficult to interpret. Therefore, we chose to present population prevalence by age categories based on different social and physical developmental stages for adolescents.

3. Table 1 - data is presented for "Adolescents whose entire ten years of adolescence (ages 10-19) occurred in 1998-2011" which makes sense. But what about the denominator? Is the cited 3,254,046 also for all adolescents 1998-2011? Also, surely you would need births and deaths data to approach an accurate figure?

Response:

This is correct, the denominators are ONS mid-year population estimates for all adolescents from 1998-2011. These estimates have already taken births and death rates into account and assumed even rates of immigration and migration out of England. More details can be found here: <http://www.ons.gov.uk/ons/rel/pop-estimate/population-estimates-for-uk--england-and-wales--scotland-and-northern-ireland/a-short-guide-to-population-estimates/index.html>

4. It would be nice to know the distribution of admissions, that is how many had one AdvRI, two, three... and so on. At the moment Table 2 stops at "3+". Might be nice to have cumulative admissions in a graph for each group.

Response:

We did examine the distribution of the number of admissions as part of internal analyses, and found a small proportion of adolescents had more than 3 admissions during their adolescence (0.8%). We repeated the analyses for 1, 2, 3, 4+ emergency admissions for injury and admissions of any type, and found that the conclusion of a higher proportion of adolescents with 2, 3, or 4+ in the adversity group compared to other groups, did not change. We would prefer to keep the admissions presented as in the current Table 2 as not to lose the main message, which is that both emergency admissions for injury and admissions of any type are most frequent in the Adversity group.

5. This paper focuses only on admissions the very severe end of the spectrum. A great number of youngsters will attend, be treated but not admitted. It is important that this paper acknowledges this. While not as costly there are still impacts on services. Particularly if making the case that interventions are required in this group.

Response:

Unfortunately data on accident and emergency attendances in England are not as well coded as for admissions,(5) and are not available before 2007.

Minor comments:

6. Page 2 - line 40. Numbers are hard to read

Response:

Thank you. We have now amended this.

7. Supplementary Table 1 does not seem to add very much

Response:

We would prefer to keep this table because it demonstrated the characteristics of individuals in our study and calendar-time periods their adolescence covered. This table assists readers, such as practitioners and researchers, to understand how the cohort and denominators were derived.

Reviewer: 3

Reviewer Name Peter Sidebotham

Institution and Country Warwick Medical School, UK

This is a tidy study using secondary national data to examine adversity-related injury admissions among adolescents in England. The messages are straightforward and important:

- adversity-related injury is prevalent in the English adolescent population, accounting for 60% of all emergency admissions for injury in girls and 30% in boys;
- co-morbidity is common, with many of this group suffering multiple adversities;
- recurrence is more common in this group than among those with accident-related injury.

These are extremely important messages with implications for practitioners and policy makers/public health.

The study has been well designed, and the authors have been thorough in describing their approach to coding adversity-related injury. The methods are clearly described and appropriate.

1. I was uncertain that the inclusion of coding for and analysing the data in relation to chronic conditions really added anything to the paper. Unless you are making any hypotheses in relation to

chronic conditions, or intend to explore this in depth, I suspect it adds an unnecessary layer of complexity. I would exclude this component of the analysis in the interests of keeping the key messages clear and simple.

Response:

We have now removed the rows for chronic conditions from Table 1 and briefly state the proportion in each group who has a chronic condition in the Results section.

The results are clearly presented and easy to follow. The tables are clear (as above, the row on chronic conditions could be omitted).

The discussion is well presented, thorough and appropriate in the light of your results and the wider literature. You discuss the limitations of this dataset. You draw out appropriate policy implications.

2. I felt figure 1 was unnecessarily complex. Why not just present as 2 straightforward Venn Diagrams? The data in the boxes on the right could simply be incorporated in the relevant fields of the Venn diagram.

Response:

We have now presented these data more simply as proportional Venn diagrams.

Minor comments:

4. p6, line 93 - you need to be consistent in hyphenating adversity-related

p6, line 95 - need to know

3. p8, line 128 - as being related to

Response:

Thanks for spotting these grammatical errors. These have now all been corrected in the revised manuscript.

5. p9, line 134 - I presume the comment 'and excluded 'undetermined intent'.' means this was excluded from the self-harm category, but included under assault. This could be clarified, as my first thought on reading it was that you were excluding it entirely, which would not be appropriate as it implies some adversity.

Response:

That is correct. Since submitting the manuscript we have noticed that in ICD-10, none of the codes which have the phrase 'self-harm' or 'self-poisoning' also have the phrase 'undetermined intent'. Therefore, we have now removed the sentence you refer to from the main text.

6. p16, line 233 - adolescents who were exposed to

Response:

Thanks, this has been corrected.

7. In the supplementary table (p33, line 28), surely neonatal withdrawal symptoms could be excluded as have other neonatal conditions.

Response:

See our response to Reviewer 1 comments point 3.

8. p36, line 34 - should X65 be included in both SH and DA? Similarly X66 (p38 line 5) 8. p38, line 29

- I am unconvinced by the exclusion of Z86.4 because it includes tobacco. Surely the inclusion of alcohol and drugs here makes its inclusion important? This may need further justification.

Response:

X65 is only considered as self-harm in this study. We included it as part of the drug or alcohol misuse list to show that it would have been considered had it not been self-harm already, as these clusters may be used in future for work in drug or alcohol misuse in adolescents where self-harm is not also being studied. This is the same case for X66. We have no way of separating out Z86.4 to know how much of this is accounted for by drugs or alcohol, and so it would not be a very specific code to use for the DA cluster. However, we did check how many adolescents who were not already in the DA cluster had this code and the number was very small ($n = 405$; the DA cluster currently contains 107,659 individuals). Therefore, excluding Z86.4 from the definition of drug or alcohol misuse should not substantially affect results or conclusions in this study.

9. p40, line 6. In the title for this table, you need to specify that the population prevalence is per 100,000 (I presume)

Response:

Thanks for spotting this. The figures were originally per 1,000, but we now present them per 100,000 and indicate this in the title.

Reviewer: 4

Reviewer Name Kirsten Vallmuur

Institution and Country Queensland University of Technology

Centre for Accident Research and Road Safety Queensland

This paper describes admissions for adversity-related problems and injury in adolescents (10-19 years) in England using longitudinally linked retrospective hospital admission data in England. It is a very well written, innovative and interesting paper using a well constructed methodology, which addresses a significant problem of concern to the health sector and larger community. The comments I have are all focused on providing greater clarity in the description to assist the readers understanding, as well as a few minor editorial suggestions.

1. Firstly, my most substantial recommendation which affects the wording used throughout the paper is around the use of the term 'injury'. While I note that the Supplementary Table 2 states that for a case to be considered an injury it would have an S or T code from the ICD in the first episode of admission, this is the only place where this restriction is indicated; the paper itself refers the reader to the adversity codes and accident codes on page 8 line 128-129 "we defined an emergency admission for injury as being for related to adversity....." (note the grammar needs correcting in this sentence on review), and hence it is unclear to me whether the case cohort is required to have S or T codes to be included OR whether it can have S or T codes or any of the adversity codes. This needs to be made clearer and if it is the latter (ie S OR T OR Adversity codes) I have some thoughts on the terminology used throughout the paper. If using the operational definition of 'injury' which is largely employed in the injury field, technically injuries are only those coded to Chapter 19 and/or Chapter 20 of the ICD-10. Even the more theoretical definitions of injury which have been put forward over the years (such as the World Health Organisation definition in Holder Y, Peden M, Krug E et al (Eds). Injury surveillance guidelines. Geneva, World Health Organization, 2001) would not cover the entirety of the concepts which are included within the 'adversity' category. I agree with the breadth of the adversity concept, which I think is an inclusive and comprehensive list of relevant codes, however I think that there could be a better use of terminology than 'injury' in relation to the proposed 'adversity' category. One approach could be to use the phrases:

- ‘adversity-related injuries and social problems’ (or simply ‘adversity-related problems’) instead of ‘adversity-related injuries’;
- ‘adversity and/or injury emergency admissions’ instead of ‘emergency admissions for injury’; and
- ‘problems related to adversity’ instead of ‘injury related to adversity’.

I agree with the use of the term ‘accident-related injury’ for the categories referring to the cases with a code in the unintentional injury code range without the adversity codes, as these are within the scope of what is an ‘injury’.

Response:

Adolescents were only included in the cohort if they had ever had an emergency admission for injury (the former rather than the latter option given: “the case cohort is required to have S or T codes”). Those who were in the cohort were then grouped according to whether their emergency admissions for injury were also related to adversity, or not adversity but accidents, leaving a final ‘other’ group. On lines 116-117 we state that each individual in the cohort must have at least one emergency admission for injury.

Thanks also for your comment about how ICD-10 is used and references. We have used Chapter 19 codes only to define injury. We understand that even if Chapter 20 was also used to define injury, not all of our codes used to define adversity are within Chapters 19 and 20 of ICD-10, and do not necessarily represent the direct cause of the injury. However, now that we have removed adverse social circumstances from the definition of victimisation (in response to Reviewer 1, comment 3), the codes listed in Supplementary Table 2 (maltreatment, assault, undetermined causes of injury, self-harm, drug or alcohol use) when present at an emergency admission for injury would be related to the presenting injury. Therefore, we would prefer to keep the phrase ‘adversity-related injury’ in the manuscript.

Thanks for spotting the grammatical error, we have now corrected this.

The other suggestions I have for improving clarity are as follows:

2. • TITLE: The title should reflect the change suggested in previous paragraph – suggested wording ‘...in England for adversity-related problems and injury.....’;

Response:

We assume that this comment only applies if we had looked at injuries or adversity rather than injuries and adversity. As discussed in the previous comment, the cohort all had S or T codes (injury) at emergency admissions, and then we investigated the proportions of the cohort who had victimisation (violence), self-harm or drug or alcohol misuse codes within these emergency admissions for injury. We feel that the current title reflects this.

3. • ABSTRACT, DESIGN: Include the fact that data longitudinal and linked as this makes it unique – i.e. ‘we used longitudinally linked hospital administrative data’

Response:

We agree this is a good point to make, and now have done so on line 34.

4. • METHODS, ADMISSION DATA: Last sentence indicates the codes for ‘other causes’ is in the table but they aren’t currently and there needs to be either a broad range provided or a statement to say ‘all other codes in ICD other than those listed above’.

Response:

We think that’s a good point. We removed ‘or other causes’ from this sentence. Other causes are now introduced a lot later under the ‘Classification of adolescents according to types of injury and age at

emergency admissions' (lines 151-152).

5. • METHODS, TYPES OF INJURY: Suggest adding to '(maltreatment/assault)' line 129 '(maltreatment/assault/social problems)' to give the reader an early indication that the adversity category is broad.

Response:

We agree and have now included this.

6. • METHODS, TYPES OF INJURY: Age ranges in line 142 are overlapping and should be (15-17 years) and (18+ years).

Response:

We have now changed this.

7. • RESULTS, FIRST PARAGRAPH: there were 402916 cases but 462476 emergency admissions? Does this include the multiple presentations for some of the cases? If so this needs to be made more explicit.

Response:

That's right. We have tried to make this clearer on line 190.

8. • RESULTS, TABLE 1: It wasn't clear till I found the footnote that the table referred to the first emergency admission and I would suggest including this in the column heading and maybe even the table title to make it more transparent.

Response:

Thanks, we take your point and have moved the footnote to be part of the title.

9. Also, I don't see the need to repeat the column headings and 'All' row again in the second row from the bottom and I would suggest just including the chronic conditions in a consecutive row after the Missing (sex) row.

Response:

We repeated the headings because the first section included row %s and the second included column %s. In response to Reviewer 2, comment 1, we have removed the chronic conditions section from this table and only mention these results in the text, as to not deflect from the main results.

10. • RESULTS, TYPES OF ADVERSITY-RELATED INJURY: Line 230, add 61.7% 'of all of the AdvRI group' to make it clear what the denominator is, and similarly for line 232 after 22.7%.

Response:

Thanks for this point. We have included this line after 61.7% and feel that it makes it clear that the following %s all have the same denominator too.

11. • RESULTS, TYPES OF ADVERSITY-RELATED INJURY: Line 233, should read 'for most of the adolescents who were exposed.'

Response:

See our response to Reviewer 3 comments point 6.

12. • RESULTS, TYPES OF ADVERSITY-RELATED INJURY: Line 237, should read 'simultaneously

in at least'.

Response:

Thanks for spotting this, this has now been changed.

13. • RESULTS, TABLE 2: Is it possible to make the columns AdvRI, AccRI, Other instead of Emergency admissions and Admission of any type? These latter categories don't mean much in the context of the article whereas having some indication of how many cases had second and third representations for adversity related problems or Injury related problems for each group using all of the same row headings would be of interest to get a more complete understanding of the trajectory of these patients.

Response:

We feel that being re-admitted with any admission reflects extra burden on the individuals and health services. Our hypothesis is that these adolescents with adversity-related injury often have underlying psychological and social need, and are more likely to be re-admitted with any problems, not just injury. We also include the burden of emergency admissions for injury, to indicate the pathways back into hospital. We are currently doing further work on what these adolescents come back in with in more detail, but feel it is not within the scope of this current paper.

14. • DISCUSSION: In regards to Line 300 where the authors indicate a higher rate of drug/alcohol misuse in girls in their study which they state is different to other studies, the study recently published online in DAR on ED injury presentations in youth and the level of drug/alcohol involvement found the same finding (See <http://www.ncbi.nlm.nih.gov/pubmed/25303680>).

Response:

A higher rate of drug or alcohol misuse for girls in hospital admissions is different to other studies of adversity in the general population. Thanks for bringing this recent paper also in a hospital population to our attention. It is interesting to see that girls who present with injury are more likely than boys to present with injuries solely caused by intoxication, but boys are still more likely to present with alcohol-related violence and falls and other types of alcohol-related injury.

15. • DISCUSSION: Line 304 – are those with adversity presentations more likely to be readmitted if they reappear even if the injury is minor so as to remove them from dangerous or suboptimal home environments (homeless, domestic violence etc) for a short stint, and does this impact on the findings in relation to readmissions?

Response:

Sorry we have no way of knowing about this without linked emergency department data.

16. • DISCUSSION: Line 315 – the authors mention there are no clinical guidelines for assault, so does this mean there are clinical guidelines for self-harm and alcohol/drug admissions? If not then these need to be added to the previous sentence in the manuscript.

Response:

There are clinical guidelines for management of patients with self-harm and drug or alcohol misuse,(6-8) which are referenced in the following sentence.

17. • DISCUSSION: Needs a concluding paragraph to tie up the paper, and this could briefly mention what the international significance of this project is (i.e. ability to replicate methods in other countries that have linked data etc).

Response:

We have now added a more conclusive paragraph (lines 326-335).

1. Viner RM, Ozer EM, Denny S, Marmot M, Resnick M, Fatusi A, et al. Adolescence and the social determinants of health. *The Lancet*.379(9826):1641-52.
2. Bellis MA, Hughes K, Wood S, Wyke S, Perkins C. National five-year examination of inequalities and trends in emergency hospital admission for violence across England. *Injury prevention : journal of the International Society for Child and Adolescent Injury Prevention*. 2011;17(5):319-25.
3. Truth hurts: report of the national inquiry into self-harm among young people. London: Mental Health Foundation; 2006.
4. Office National Statistics. Smoking, drinking and drug use among young people in England in 2011. Stationery Office, 2011.
5. Hospital Episode Statistics Accident & Emergency (England), 2007- [19th December 2014]. Available from: <http://adn.ac.uk/catalogue/cataloguepage?sn=888040>.
6. Self-harm. The short term physical and psychological management and secondary prevention of self-harm in primary and secondary care. Health Do, editor. London: National Institute for Health and Clinical Excellence; 2011.
7. Drug misuse: Psychosocial interventions. London: National Institute for Health and Clinical Excellence; 2007.
8. Alcohol-use disorder : diagnosis, assessment and management of harmful drinking and alcohol dependence. London: National Institute for Health and Care Excellence, 2011 9781849364911 (pbk.).

VERSION 2 – REVIEW

REVIEWER	Simon Moore Cardiff University, UK
REVIEW RETURNED	05-Jan-2015

GENERAL COMMENTS	I am broadly happy with the authors response to reviewers' comments and happy to see this go forward.
---

REVIEWER	Kirsten Vallmuur Centre for Accident Research and Road Safety Queensland, Australia
REVIEW RETURNED	08-Jan-2015

GENERAL COMMENTS	I believe the authors have addressed the reviewers comments well and the paper has been strengthened through the process. One remaining comment I think needs to be included in the discussion is Reviewer 2's 5th comment. The authors could include a statement acknowledging that the sample reported does represent the more severe end of the spectrum and that the impact of adversity-related presentations is larger than the numbers represented in this paper, but that the lack of accident and emergency attendance data limits the ability to fully quantify the problem. Congratulations to the authors on a well written paper and response to reviewers, and I look forward to seeing it in published form.
---

REVIEWER	Peter Sidebotham Warwick Medical School, UK
REVIEW RETURNED	13-Jan-2015

GENERAL COMMENTS	This revision has clearly addressed both my own and other reviewers' previous comments. The paper reads well. I note that in one box in supplementary figure 1, V is still referred to as victimisation rather than violence. I have no other comments to add.
--

REVIEWER	David Cottrell University of Leeds, UK
REVIEW RETURNED	20-Jan-2015

GENERAL COMMENTS	The authors have addressed the issues raised by my first review. I always thought this was an important piece of work and I would be happy to see it published now.
---

VERSION 2 – AUTHOR RESPONSE

Reviewer: 4

Reviewer Name Kirsten Vallmuur

Institution and Country Centre for Accident Research and Road Safety Queensland, Australia

Please state any competing interests or state 'None declared': None declared

I believe the authors have addressed the reviewers comments well and the paper has been strengthened through the process. One remaining comment I think needs to be included in the discussion is Reviewer 2's 5th comment. The authors could include a statement acknowledging that the sample reported does represent the more severe end of the spectrum and that the impact of adversity-related presentations is larger than the numbers represented in this paper, but that the lack of accident and emergency attendance data limits the ability to fully quantify the problem. Congratulations to the authors on a well written paper and response to reviewers, and I look forward to seeing it in published form.

Response: We have now included a statement to this effect (lines 290-296).

Reviewer: 3

Reviewer Name Peter Sidebotham

Institution and Country Warwick Medical School, UK

Please state any competing interests or state 'None declared': None declared

This revision has clearly addressed both my own and other reviewers' previous comments. The paper reads well. I note that in one box in supplementary figure 1, V is still referred to as victimisation rather than violence. I have no other comments to add.

Response: Thankyou for spotting this. This has now been corrected.